# A Review of Antimicrobial Polymer Coatings on Steel for the Food Processing Industry

**DOI:** 10.3390/polym16060809

**Published:** 2024-03-14

**Authors:** Ksenia Sukhareva, Vasily Chernetsov, Igor Burmistrov

**Affiliations:** 1Higher School of Engineering, Plekhanov Russian University of Economics, 36 Stremyanny Ln, 117997 Moscow, Russia; burmistrov.in@rea.ru; 2N.M. Emanuel Institute of Biochemical Physics, Russian Academy of Sciences, 4 Kosygin Str., 119991 Moscow, Russia; 3ORELMETALLPOLYMER LLC., 1yu Avtomagistral Street, 303032 Mtsensk, Russia; info@orelmp.ru; 4Department of Functional Nanosystems and High-Temperature Materials, National University of Science and Technology “MISIS”, 4 Leninsky Pr, 119049 Moscow, Russia

**Keywords:** antibacterial metals, antibacterial steel, antimicrobial polymer coatings, food contact material

## Abstract

This article will focus on the issue of protection against the pathogenic biofilm development on steel surfaces within the food sectors, highlighting steel’s prominence as a material choice in these areas. Pathogenic microorganism-based biofilms present significant health hazards in the food industry. Current scientific research offers a variety of solutions to the problem of protecting metal surfaces in contact with food from the growth of pathogenic microorganisms. One promising strategy to prevent bacterial growth involves applying a polymeric layer to metal surfaces, which can function as either an antiadhesive barrier or a bactericidal agent. Thus, the review aims to thoroughly examine the application of antibacterial polymer coatings on steel, a key material in contact with food, summarizing research advancements in this field. The investigation into polymer antibacterial coatings is organized into three primary categories: antimicrobial agent-releasing coatings, contact-based antimicrobial coatings, and antifouling coatings. Antibacterial properties of the studied types of coatings are determined not only by their composition, but also by the methods for applying them to metal and coating surfaces. A review of the current literature indicates that coatings based on polymers substantially enhance the antibacterial properties of metallic surfaces. Furthermore, these coatings contribute additional benefits including improved corrosion resistance, enhanced aesthetic appeal, and the provision of unique design elements.

## 1. Introduction

The capability of bacteria to attach to solid surfaces, proliferate, and form bacterial biofilms raises significant concerns across various industries, notably the food processing sector. This phenomenon contributes to surface contamination in food processing lines, ultimately compromising food safety and shelf-life [1,2,3,4]. The concave surface features, when harboring organic residues, can facilitate bacterial cell attachment and the formation of biofilms, posing a significant challenge in the food and beverage industries. Contamination and biofouling of food contact surfaces can lead to outbreaks of foodborne diseases [5,6]. Pathogens such as *Salmonella enterica*, *Listeria monocytogenes*, *Escherichia coli*, *Bacillus cereus*, *Shigella* spp., and *Vibrio* spp. stand out as some of the most common causes for contamination in food processing plants. Surfaces that do not come into direct contact with food, including machinery parts, floors, ceilings, walls, and sewage systems, can indirectly introduce contamination into food products. To mitigate this risk, it is crucial to continually prevent bacterial attachment and growth, emphasizing proactive measures over solely relying on bacteria elimination during sanitation procedures. Microbial contamination has the potential to manifest at various stages of the food production process, including production, processing, distribution, and preparation [7,8,9,10]. Biofilms have the capability to form on a broad spectrum of food contact surfaces within food industry plants such as conveyor belts, cutting tools, steel surfaces of open or closed processing equipment, rubber seals, and so on [11,12]. This contamination may occur due to interactions with multiple cycles of raw products between sanitation cycles, the processing environment, and operators. While metals find extensive use in engineering and medical applications, bacteria can attach, grow, and develop biofilms on metals and various other materials.

Steel emerges as a paramount material within the food processing sector, lauded for its extensive application across food contact surfaces. Its selection for crafting food processing and handling apparatus stems from its robust mechanical and chemical attributes, compatibility with both biological materials and foods, exceptional corrosion resistance, ease of cleaning, cost-effectiveness, and safety. Its adoption is particularly pronounced in environments where maintaining surface cleanliness is essential, spanning from domestic settings like kitchen sinks to the food industry with its processing and storage tanks and food preparation counters, extending into the medical realm for operating tables and surgical instruments. Nevertheless, steel’s intrinsic high surface energy, typical roughness, and affinity for water render it prone to biofouling through organic food residues [13,14]. The stainless-steel spectrum encompasses over 150 variants. The American Iron and Steel Institute (AISI) categorizes these into series such as the 200 series with high manganese austenitic steels, the austenitic 300 series, and the 400 and 500 series embracing ferritic and martensitic steels, respectively. Despite this variety, only a select few are deemed suitable for food-grade applications. Dominating the stainless-steel production, austenitic stainless steel, particularly the AISI 200 series (e.g., 201, 202) and the AISI 300 series (e.g., 304, 302, 316) constitutes 70–80% of the output and is favored in dairy and food-processing endeavors. This steel type is distinguished by its non-magnetic nature, ductility, resistance to hardening through heat treatment, and simplicity in fabrication. It boasts a significant chromium and nickel content, with austenitic steels typically incorporating 8–30% nickel alongside varying molybdenum levels, albeit vulnerable to chloride stress corrosion cracking above 55 °C. Among these, the AISI 304 grade stands out for its approximately 0.05% carbon, 18% chromium, and at least 8% nickel composition, presenting a cost-effective choice for a wide range of food processing settings. Renowned for its compatibility with various food processing requirements, AISI 304 stainless steel is celebrated for its chemical and physical stability, high corrosion resistance, and ease of sanitation. It is highly workable, supporting fabrication into diverse equipment forms, and is commonly employed in manufacturing storage tanks, brewing vessels, kitchen sinks, fermentation vats, milk tankers, dishwashers, and more. Meanwhile, AISI 316 stainless steel, enriched with 2–3% molybdenum, excels in high-temperature resilience, crucial for welding applications, and showcases formidable resistance to a broad spectrum of chemicals including chlorides, alkalis, and acids. This makes it an ideal candidate for constructing plate heat exchangers, CIP tanks, and evaporator tubes requiring enhanced corrosion resistance. The low carbon variants, 304L and 316L, with a carbon content of 0.03%, enhance weldability for thicker stainless-steel sections. In scenarios involving acidic fluid foods or those containing SO_2_, the preference leans towards AISI 316 or 316L over 302 or 304 for their superior performance. Lower-grade austenitic stainless steels, such as those in the AISI 100 and 200 series, are generally not advised for dairy and food equipment usage. In contrast, ferritic stainless steel (AISI 400 series, e.g., 410, 430), characterized by its magnetism and heat-treatable nature, finds utility in highly oxidative contexts due to its commendable corrosion resistance. These steels, also known as ‘ferritics,’ are optimal for non-aggressive environments where wear resistance is a priority. AISI 430 grade, with its adequate corrosion resistance and cost efficiency, is frequently chosen for applications involving prolonged exposure to mildly acidic substances, demonstrating resilience against oxidation, sulfur, and corrosion cracking [15].

Current sanitation practices in food processing facilities rely on the use of chemical disinfectants, including hypochlorites, quaternary ammonium compounds (QAC), hydrogen peroxide, and carboxylic acid. The primary approach to prevent biofilm formation involves the regular cleaning and disinfection of surfaces. However, disinfectants can prove ineffective due to biofilm resistance to common chemical and physical sanitation treatments, making their removal challenging [16,17]. Furthermore, many sanitizing/cleaning agents often contain chloramines or hypochlorites, which can damage the passivating oxide of steel, leading to corrosion. Gibson et al. conducted tests on the resistance of biofilms to sanitizers and found that a commercial alkaline detergent and an acidic cleaner were ineffective against *Pseudomonas aeruginosa* and *Staphylococcus aureus* biofilms on steel surfaces [18]. Pan et al. observed that a commercial hydrogen peroxide-based agent was ineffective in eliminating *Listeria monocytogenes* biofilms, and the bacteria exhibited greater resistance to the disinfectant based on quaternary ammonium compounds (QAC) [19].

Bacteria residing in microstructured grooves, cracks, or scratches pose a challenge for disinfectants to reach, leading to difficulties in sanitizing ‘hard to clean’ parts of food processing equipment and resulting in poor or incomplete sanitization. Consequently, the development of preventive mechanisms to mitigate contamination and biofouling on steel surfaces becomes crucial for ensuring the safety of food products [20]. Referred to as indirect food additives, food contact materials are one of the factors of food product safety, as was detailed in [4,21]. Modifying surface properties emerges as a strategy to prevent or limit bacterial attachment and reduce biofilm formation [22]. A valuable approach to enhance food safety involves inhibiting the initial bacterial attachment, thereby preventing biofilm formation on food processing surfaces. Protective coatings, acting as barriers between environments and various substrates, represent the most commonly adopted solutions across industries to counteract biofilm formation. In recent years, techniques have been developed to create uniform, robustly bonded, and antibacterial polymer surfaces on metallic substrates. In addition to their use in the food industry, antibacterial coatings find significant application in the treatment of steel alloys. This strategy seeks to imbue civil infrastructure elements like handrails and surfaces with antimicrobial features. Such enhancements are particularly valuable in environments that experience heavy foot traffic or serve as durable, biocompatible constructs in public spaces including transit systems, shopping centers, healthcare facilities, educational institutions, and more [23].

Antimicrobial polymer coatings are renowned for their effectiveness in combatting the proliferation of bacteria on steel surfaces [24]. Recent trends in the global food grade metal coating market reflect an increasing demand for eco-friendly and sustainable coatings, aligning with consumer preferences for healthier and environmentally conscious products. Traditional polymers employed as internal surface protective coatings for food preservation and metal substrate corrosion protection include polyurethane, silicone, epoxy resins [25], coal-tar epoxy, polyvinyl chloride, polyimides, and fluorinated compounds such as polytetrafluoroethylene (PTFE), polyvinylidene fluoride (PVDF), ethylene chlorotrifluoroethylene (ECTFE), perfluoroalkoxy alkane (PFA), and fluorinated perfluoroethylenepropylene (FEP) [26].

Polymer-coated steels exhibit high abrasion and corrosion resistance, coupled with exceptional appearance and moisture barrier properties. The additional protection against chemical corrosion provided by polymer antimicrobial coatings allows for significant cost efficiencies due to the ability to use ferrous steel instead of more expensive steels. In response to the evolving landscape, the food industry is witnessing the development of advanced antimicrobial metal coating technologies. These technologies effectively inhibit bacterial growth on surfaces, ensuring food safety and prolonging product shelf life.

## 2. Treatment Strategies for the Prevention of Bacterial Adhesion and Biofilm Formation on Steel for Food Application

Various approaches to control and prevent the colonization and growth of biofilm on steel have been discussed in the literature:Steel composition (alloying) [27,28,29,30];Electrochemical, chemical, or physical modification of the steel surface: metal coatings (Cu-coated steel [31,32] Ag-coated steel [33,34], Cu-Co-coated steel [34]) laser irradiation [35], etching and plasma [36,37,38,39] or other introduction of modifiers into the surface;Application of antimicrobial coatings [40,41,42,43,44].

Three primary directions stand out for antimicrobial polymer coating strategies [4]:Coating the steel surface with released-based antimicrobial coating;Coating the steel surface with contact-based antimicrobial coating;Immobilizing antifouling coatings on the steel surface to prevent bacterial adhesion.

Antibacterial surfaces can be achieved either by loading coatings with antibacterial agents (released-based surfaces) or by chemically grafting biocides onto the surfaces (contact-based antimicrobial coating). In the first case, antibacterial activity results from the diffusion of biocides out of the coating. In the second case, bacteria are killed upon contact with the surface. Surfaces designed to release antimicrobial agents not only obstruct the adherence and proliferation of bacteria but also neutralize bacterial cells. Yet, a significant limitation of such materials lies in their finite period of effectiveness. The effectiveness of antibacterial surfaces on steel surfaces vary under different conditions and depend on the nature of the surface and the surrounding environment [45]. These directions for antimicrobial polymer coating strategies on the steel surface are illustrated in Figure 1.

### 2.1. Released-Based Antimicrobial Polymer Coatings

The release-based antibacterial coatings on the steel substrate are created by integrating antibacterial agents into the coating. In this approach, the antibacterial molecules are embedded within a polymer matrix and are gradually released from the surface through processes like diffusion or erosion [4,46]. Embedding bactericidal compounds into a coating that gradually dispenses its active ingredients is a widely adopted method for altering surfaces that come into contact with food to regulate bacterial proliferation. Polymer surfaces on steel, engineered for controlled release, not only prevent bacteria from settling and multiplying but also render bacterial cells inactive. Commonly used active antibacterial agents include metal nanoparticles (NPs) which can release active ions or diffuse to the polymer surface, essential oils, antibiotics, etc. However, a notable drawback of these materials is their limited effective lifetime. Since antibacterial agents are loaded into the system at a fixed amount, the coatings become ineffective once the agents are completely released [4,46,47].

#### 2.1.1. Antibacterial Polymer Coatings Functionalized with Metal Nanoparticles

Antibacterial polymer coatings for steel surfaces can be significantly improved by incorporating metal nanoparticles and derivatives, such as metal oxides, into their structure. Metal nanoparticles actively combat bacterial colonization by repelling bacterial adhesion, killing adherent bacteria, or inhibiting biofilm formation [48]. Numerous studies have demonstrated that metals and their oxides can be directly grafted onto the steel substrate using various methods such as electroless processes, electrochemical deposition, and atomic layer deposition (ALD) techniques [23,40,49,50,51,52,53] or be incorporated as antibacterial additives in polymer coating for steel. Table 1 summarizes some recent efforts regarding the development of antibacterial polymer coatings functionalized with metal NPs.

Silver nanoparticle (AgNP)-based coatings are considered one of the most promising types of metal nanoparticles due to their broad-spectrum activity against various pathogens, including microbes, fungi, bacteria, and viruses, even in small amounts. The antimicrobial efficacy of silver is approximately 100 times stronger than that of copper [54,55]. However, the metallurgical process of Ag-containing antibacterial steel is limited, resulting in an uneven distribution of the antibacterial phase and a loss of antibacterial properties [56,57,58].

While commercially available silver-based antimicrobial coatings exist, they face limitations such as high cost and ineffectiveness in food manufacturing environments due to fouling from organic loads and antimicrobial resistance. Composite coatings, combining antibacterial AgNPs with a non-stick PTFE coating, are frequently described in the literature as a type of polymeric coating for steel, providing enhanced resistance to bacterial contamination.

Various approaches have been proposed for forming PTFE-based coatings on steel substrates. In a study by Zhao et al. the electrolysis method was applied to prepare Ag–PTFE composite coatings on 316L steel, exhibiting antimicrobial and anti-corrosion properties [59]. The Ag-PTFE coated steel reduced *Escherichia coli* attachment by 94–98% compared to silver-coated steel, steel 316L sheet, or titanium sheet. Another study by Zhang et al. utilized a sol-gel-based dip coating method to coat the polytetrafluoroethylene (PTFE) layer on steel, followed by immobilizing AgNPs on the surface, resulting in an antibacterial nanocomposite structure on steel [60]. This structure exhibited prolonged antibacterial activity against *Escherichia coli* and enhanced corrosion resistance [60]. Karabulut et al. described the coating containing silver nanoparticles (AgNPs) based on Locust bean gum (Loc) and polyethylene glycol (PEG) [61]. Loc is a natural polymer and is widely employed in various industries, particularly in the food and pharmaceutical sectors. In this study, Loc and PEG were utilized as stabilizing and reducing agents for the AgNPs. The resulting Loc/PEG-AgNPs were applied to the steel surface through drop casting and airbrush spray coating techniques. The technique of integrating diverse methods is explored, highlighting the combination of silver nanoparticles with the cationic polymer poly(3,4-dihydroxy-L-phenylalanine)-co-poly(2-(methacryloxy)ethyl trimethylammonium chloride) (DOPA) to bolster the adherence of a sequential deposition coating on steel surfaces. DOPA possesses the capability to attach to a variety of inorganic substrates, steel included. This concoction yields a stable aqueous dispersion of Ag^0^ and AgCl nanoparticles, which is then merged with the polyanion poly(styrene sulfonate) (PSS) atop the steel. The positive charge of DOPA and the negative charge of PSS interact, leading to micelle formation. Such a layered (LbL) coating demonstrates potent antibacterial properties, primarily due to the silver ions disseminating from the film, especially effective against strains of *Escherichia coli*. Applying a layer-by-layer technique to coat steel with a suspension of P(DOPA)-co-P(DMAEMA+)/AgCl/Ag^0^ in water, followed by a water-based solution of polystyrene sulfonate, bestows robust antibacterial properties against Gram-negative *Escherichia coli* bacteria. The presence of DOPA units ensures strong anchoring to the steel substrate, while the silver nanoparticles serve as sources of biocidal Ag+, providing antimicrobial activity against Gram-negative *Escherichia coli* bacteria. The films can be reloaded with AgCl by simply dipping them in an aqueous AgNO_3_ solution, thereby enhancing antibacterial activity once again. The entire film formation process, including copolymer synthesis, is carried out in aqueous media under very mild conditions, making it highly attractive for industrial scale-up and sustainable applications [62]. In the Pereyra et al. study, various samples of NaA were prepared through cation exchange, each containing different amounts of Ag^+1^ and Zn^+2^ [63]. These samples were created to assess their antibacterial properties. The introduction of AgZnA into the epoxy matrix leads to a reduction in the number of bacteria adhering to the coating, consequently lowering the corrosive impact caused by *Pseudomonas aeruginosa* [63].

Cowan et al. assessed the antibacterial effectiveness of steel specimens treated with a zeolite matrix imbued with silver and zinc, referred to as AgION [64]. These coated surfaces exhibited antibacterial capabilities against various bacterial strains including Gram-positive organisms such as *Listeria monocytogenes* and *Staphylococcus aureus* as well as Gram-negative bacteria such as *Pseudomonas aeruginosa* and *Escherichia coli.* [64]. Pishbin et al. developed composite antibacterial coatings for steel, utilizing a Bioactive Glass/Chitosan/Nano-Silver matrix via electrophoretic deposition. These layers demonstrated exceptional antibacterial performance, particularly targeting *Staphylococcus aureus* [65]. Liu et al. developed a unique coating system utilizing poly(lactic-co-glycolic) acid (PLGA) as a biodegradable carrier for enclosing silver nanoparticles (Ag-NPs) [64]. This cutting-edge approach entailed a threefold immersion of steel into a chloroform solution containing 17.5% PLGA (by weight/volume), combined with spherical Ag-NPs ranging from 20 to 40 nm in size. Each immersion lasted 30 s, and was followed by a 12 h incubation at 37 °C. Remarkably, a coating with 2% silver content in the PLGA matrix demonstrated not only an inhibitory effect on the growth of pathogens like *Staphylococcus aureus* and *Pseudomonas aeruginosa* in a laboratory setting but also, when tested in vivo using a rat femoral canal model, showed complete absence of bacterial presence near the implant after a period of eight weeks.

In their study, Qian et al. developed a multi-layered coating for stainless steel grade 316L involving polydopamine (PDA) and silver nanoparticles (AgNPs) [66]. This innovative approach was aimed at endowing the material with exceptional characteristics, including superhydrophilicity, antibacterial capabilities, and enhanced resistance to corrosion [66]. Further research [67] examined a nanocomposite layer comprising polydopamine, modified poly(3,4-ethylenedioxythiophene) (PEDOT), and AgNPs, demonstrating the potential of such coatings.

A novel antibacterial coating system using CuNP-loaded PEGDA hydrogel has been proposed for various metallic devices susceptible to microbial contamination, particularly those made of steel. These nanostructured coatings were created through two distinct loading methods: the incorporation of CuNPs during PEGDA electropolymerization or after it. Assessing the antimicrobial efficacy of CuNPs-PEGDA coatings on steel sheets against *Staphylococcus aureus* and *Escherichia coli*, it became evident that the inhibition of bacterial growth depends on the loading method. PEGDA hydrogel coatings modified by the addition of pre-electrosynthesized CuNPs, following electropolymerization (referred to as PEGDA-CuNPs AE systems), exhibited a notable inhibitory effect against both tested microorganisms [68].

**Table 1 polymers-16-00809-t001:** Released-based antimicrobial polymer coatings with metal NPs on steel.

Antibacterial Additive	Type of Polymer Matrix	Coating Method	Properties	Reference
AgNPs	Organic Locust Gum/ Polyethylene glycol	Drop casting and airbrush spray coating	- Antioxidant capacities from 17.90 ± 0.50 to 20.47 ± 0.19 mmol Trolox equivalent (TE)g - Improved corrosion resistance - Uniform coating	[61]
AgNPs	PTFE	Electroless	- Reduced *Escherichia coli* attachment by 94–98% - Anticorrosion resistance	[59]
AgNPs	PTFE	Sol-gel-based dip coating method	- Prolonged antibacterial activity against *Escherichia coli* - Enhanced corrosion resistance in PBS	[60]
AgNPs	DOPA	Layer-by-layer deposition	- Reloading antibacterial activity	[62]
Ag/Zn-exchanged zeolite	Epoxy resin	Airless spray	- Inhibited the growth of *Pseudomonas aeruginosa* for concentrations up to 200 mg·L^−1^	[63]
AgNPs Bioactive Glass	Chitosan	electrophoretic deposition	- antibacterial activity against *Staphylococcus aureus*	[65]
CuNPs	PEGDA hydrogel	electrochemical polymerization	- Hydrophilic coating - Significant inhibitory effect against *Staphylococcus aureus* and *Escherichia coli*	[68]

#### 2.1.2. Antibacterial Polymer Coatings Functionalized with Antibacterial Enzymes

Enzymes are widely utilized in the formulation of detergents, industrial procedures, and the food sector. Recognized for their non-toxic and bioactive antifouling properties, enzymes have garnered attention as a promising source for the development of antimicrobial surface coatings [69]. In terms of bacterial adhesion, enzymes can disrupt the bacteria’s adhesion mechanisms to surfaces or catalyze the hydrolysis of peptidoglycan, leading to the disintegration of bacterial cell walls and subsequent bacterial eradication. For instance, lysozyme, a member of the hydrolase enzyme family, damages bacterial cell walls through hydrolysis of 1,4-β-linkages in peptidoglycan and chitodextrins. This process increases the cell’s permeability, eventually causing it to burst. The bacteriostatic or bactericidal impact of enzymes hinges on their utilized concentrations. Implementing lysozyme and/or poly(ethylene glycol) (PEG) on steel surfaces that have been primed with poly(ethylene imine) (PEI) has been proven to offer resistance against the attachment of proteins and bacteria. This strategy has been effective in inhibiting microbial growth of *Listeria ivanovii* and *Micrococcus luteus* [70]. Moreover, employing the serine protease enzyme trypsin to deter biofilm formation has been recorded [71]. A coating infused with trypsin on steel surfaces demonstrated significant antimicrobial prowess against *Staphylococcus epidermidis*.

In their research, Yuan et al. utilized poly(ethylene glycol) monomethacrylate (PEGMA) and lysozyme through a ‘grafting from’ approach [72]. This process included the coating of steel surfaces with a dopamine-mediated layer that served as a foundation for an alkyl halide initiator. This was followed by the initiation of surface-based atom transfer radical polymerization (ATRP) using PEG-monomethacrylate. Lysozyme molecules were then conjugated to the terminal ends of the PEG chains, employing 1,1′-carbonyldiimidazole as a biochemical linker. The modified steel surfaces proved effective in preventing bovine serum albumin (BSA) adsorption and reducing bacterial adhesion and biofilm formation. These surfaces also displayed robust bactericidal effects against *Escherichia coli* and *Staphylococcus aureus*. The combined integration of hydrophilic antifouling brushes and antibacterial enzymes or peptides on metal surfaces, using catecholic anchors, presents a versatile approach that is adaptable to various metal substrates. This strategy holds significant potential for applications in biomedicine and biomaterials. However, despite their effectiveness, the process of enzyme extraction and purification before use incurs substantial economic costs [73].

### 2.2. Contact-Based Antimicrobial Polymer Coatings

Instead of relying on a release-based approach, an alternative method involves the creation of molecular layers where antibacterial molecules are covalently immobilized on the surface, or the introduction of an antimicrobial additive, that is stable over the entire service life of the coating. This approach can be highly effective in preventing bacterial colonization. Typically, it leads to antibacterial properties that last longer and addresses concerns related to the potential adverse effects associated with the leaching of antibacterial agents. Various types of molecules can be covalently attached to surfaces to provide bactericidal activity. These include antimicrobial peptides [74,75,76,77,78,79,80], chloropolymers [20,81,82,83,84], photocatalytically active semiconductors [85,86,87], and cationic polymers [88,89,90,91].

#### 2.2.1. Antimicrobial Cationic Polymers and Peptides Coatings

Polyethyleneimine (PEI) is a polyamine polymer characterized by an abundance of primary, secondary, and tertiary amine groups. Its highly branched cationic structure enables strong adhesion to substrates, making it ideal for creating protective layers that seal surface flaws and guard against corrosion [48,92,93]. Gibney and colleagues investigated the antibacterial properties of PEI, discovering its effectiveness against both Gram-negative and Gram-positive bacteria [88]. The bactericidal effect of N-alkyl-PEI is due to its ability to disrupt bacterial cell membranes [89]. In another application, a multifunctional composite coating was developed for Mg AZ31 magnesium alloys. This coating was created using a micro-arc oxidation process that incorporated silver nanoparticles (AgNPs) and PEI. The PEI served as a matrix for evenly distributing AgNPs, forming an antimicrobial barrier against *Staphylococcus aureus* [90]. Notably, the zirconium-PEI layer displayed superior anti-corrosion properties. An innovative antifouling composite coating combining dopamine (DA), PEI, and silica (SiO_2_) was formulated and applied to 304 steel. This coating was evaluated for its antibiofilm and antibacterial effectiveness using *Vibrio natriegens*. It was found that the DA/PEI/SiO_2_ modified surface on steel achieved a 51.4% reduction in biofilm formation and a 95.2% antibacterial rate. The combined effect of DA, PEI, and SiO_2_ significantly enhanced the antimicrobial characteristics of the steel surface, while also maintaining excellent stability [91]. This synergy indicates a promising approach for enhancing the antimicrobial properties of various surfaces.

Employing antimicrobial peptides for the modification of surfaces is an innovative and effective strategy to combat bacterial contamination through the elimination of surface-bound microbes. These peptides can be anchored onto rigid platforms like steel, creating interfaces that possess lethal activity upon microbial contact [74]. Coatings derived from antimicrobial peptides demand particular consideration for their advantageous attributes, such as minimal toxicity and high safety profile, alongside superior performance relative to traditional antimicrobial agents [75]. They stand out from conventional antibiotics by offering a wide range of antibacterial, antifungal, and antiviral activities [94,95]. Furthermore, these peptides are distinguished by their robust microbe-eliminating efficiency, achieving rapid microbial inactivation at minimal effective concentrations. Crucially, they maintain their activity against strains resistant to standard antibiotics and can enhance the effectiveness of traditional antibiotics in neutralizing endotoxins. Additionally, these antimicrobial peptides are less prone to inducing bacterial drug resistance when compared to conventional biocides [76]. The immobilization of antimicrobial peptides onto a steel substrate can be achieved through various methods, one of which involves incorporating them into a polymer matrix to create a composite antibacterial coating on the surface. Numerous studies have reported the immobilization of certain antimicrobial peptides onto steel surfaces, achieved by coupling them with chitosan and other binding agents to confer antibacterial properties [78,96]. Among these peptides, one of the most commonly utilized is nisin. Héquet et al. undertook research focused on the covalent attachment of antimicrobial peptides, such as magainin I and nisin, to steel surfaces pre-treated with a chitosan polymer layer. The findings from this study demonstrated a decrease in the adhesion of *Listeria ivanovii* on the altered steel surface, underscoring the modified material’s potent anti-biofilm properties [79]. In a particular study [97], antibacterial coatings were developed through a plasma polymerization process that involved bonding allyl glycidyl ether monomers to steel. Subsequently, nisin, Tritrpticin (Trp11), or Palmitoyl-4K (4K-C16) was attached to these surfaces. The biocidal efficacy of these coatings was confirmed by achieving reductions of three to six log10 in the counts of both Gram-positive and Gram-negative bacteria in comparison to uncoated steel. Notably, surfaces treated with Trp11 demonstrated a remarkable 6.0-log CFU reduction in Gram-negative *Escherichia coli* populations compared to the control, although they did not exhibit significant deactivation of Gram-positive *Bacillus subtilis.* Surface modification of steel with antimicrobial peptides like MAG II through covalent binding has been described as an effective approach to inhibit bacterial colonization [98]. The antibacterial activity of the coating was assessed by measuring the percentage decrease (PD) in the amount of biofilm formation on the sample surface and the reduction in bacterial adhesion to the modified SS. The modified steel exhibited a PD of 71.4% against *Staphylococcus aureus* and 53.85% against *Escherichia coli*, demonstrating its antibacterial efficacy. Faure et al. demonstrated through the layer-by-layer (LbL) method that nisin can be integrated into a cross-linked coating, exhibiting enduring and potent antimicrobial efficacy against *Bacillus subtilis*. The effectiveness of these coatings was enhanced through a series of substrate immersions. Initially, a polycationic copolymer solution was firmly bound to the surface. This was followed by consecutive immersions of the surface in a poly(methacrylamide) solution containing oxidized poly(3,4-dihydroxyphenylalanine) groups, and subsequently, in a solution of a polymer enriched with primary amine groups [80]. Some of the recent advances in the development of the released-based antimicrobial polymer coatings with antimicrobial peptides on the steel are summarized in Table 2.

#### 2.2.2. Chloropolymers (N-Halamine) Coatings on the Steel

N-halamines represent a class of antimicrobial polymers that are activated upon contact, featuring nitrogen-halogen bonds created by halogenation of nitrogen-hydrogen bonds [82]. These polymers work similarly to other chlorine-based antimicrobials, releasing oxidative halogens, like hypochlorous acid, from their structure [99,100,101]. When N-halamines encounter bacterial membranes, they transfer their oxidative chloro-groups to the cells, causing bacterial destruction [102]. The application of N-halamine-based polymers on various surfaces, including steel, has gained attention for its efficacy in creating antimicrobial interfaces. This method is favored for its quick action, broad-spectrum effectiveness, cost-efficiency, and the ability to be recharged [81,103,104]. Unique to N-halamine polymers is their capacity to be re-halogenated, regaining antimicrobial strength when exposed to chlorine sources like bleach. This “rechargeability” means their effectiveness can be sustained over time using common chlorine-based sanitizers, a standard in food preparation hygiene. These polymers have shown great promise as antimicrobial paints and coatings on diverse substrates, especially steel [77,81,104]. Incorporating antimicrobial agents such as monochloramine into food contact materials requires adhering to biocide regulations, ensuring the active substances are approved for specific uses [105]. A study innovated a multifunctional N-halamine and Polypyrrole (PPy)-based coating with both electrical and antimicrobial capabilities [83]. The PPy transformed into N-halamine upon chlorine bleach exposure, displaying remarkable antimicrobial action by inactivating over 6-log CFU of *Staphylococcus aureus* and *Escherichia coli* O157:H7 in just one minute. The coating’s stability was notable, maintaining 50% functionality even after a week under fluorescent light. Further experiments used various bleach concentrations to transform PPy into N-halamines on tape, leading to the creation of efficient antimicrobial coatings on steel through electrochemical deposition. This innovative technique offers a promising avenue for developing effective antimicrobial surfaces. Demir et al. discovered that modifying steel surfaces with N-halamine-based copolymers led to a notable 6-log decrease in microbial counts, achieving total elimination within 15 min of exposure [84]. The process involved covalently attaching the copolymer to the steel, resulting in a surface with robust antibacterial properties, enduring stability, and resistance to both washing and UVA exposure. This modified steel demonstrated its antimicrobial prowess by significantly reducing *Staphylococcus aureus* and *Escherichia coli* O157: H7 populations within 15 min. Separately, Doh et al. developed an innovative food-grade hydrogel antimicrobial paint by blending gelatin with tannic acid [20]. This coating, with 1.25 ± 0.05 μmol/cm^2^ of bound chlorine, effectively combated biofouling on steel, resisting *Listeria innocua* and *Escherichia coli* O157:H7 even under extended exposure [20]. It also markedly lowered cross-contamination risks on steel when in contact with tainted produce. In essence, this antimicrobial hydrogel paint represents a promising strategy for bolstering food safety and minimizing cross-contamination risks. In the study [104], researchers developed a N-halamine and dopamine-based polymer coating with antimicrobial and adhesive properties, suitable for application on food equipment via spray-coating. The coating on steel effectively deactivated over 6 log10 CFU of both Gram-positive and Gram-negative bacteria in 10 min. Even after three discharge-recharge cycles, its chlorine levels remained high, preserving its bactericidal efficacy. This material shows promise as a high-performance, cost-effective, and easy-to-apply solution for food preparation surfaces.

#### 2.2.3. Coatings with Photocatalytically Active Semiconductors (TiO_2_) on the Steel

Steel surfaces can be enhanced with antimicrobial properties through light-sensitive compounds such as titanium dioxide (TiO_2_) and benzophenone [106,107]. These substances exhibit antimicrobial effectiveness when exposed to certain light wavelengths. TiO_2_ is renowned for its antibacterial capabilities under UVA illumination and is frequently used in self-cleaning and sterilizing surface coatings [108]. The FDA has sanctioned TiO_2_’s use in various consumer products, including food and cosmetics [85]. It can be directly applied as a TiO_2_ film on steel or incorporated as an antibacterial agent in polymer coatings [106,107,109,110,111,112,113,114]. Hung and Yemmireddy have evaluated the durability of antimicrobial coatings for food-contact surfaces [115]. They experimented with TiO_2_ nanoparticles combined with various polymeric binders on steel, using a range of organic and inorganic binders like polyvinyl alcohol, polyethylene glycol, and polyurethane. Notably, polyurethane, polycrylic, and shellac resin showed greater physical stability in TiO_2_ coatings at specific nanoparticle-to-binder weight ratios. These coatings achieved a significant bacterial reduction on the steel surfaces. Torres Dominguez et al. explored the development of mechanically robust and nanoporous TiO_2_ coatings, which produce reactive oxygen species that damage bacterial membranes and DNA, leading to their destruction [86]. Zhang et al. created a TiO_2_-PTFE nanocomposite coating for 316L steel, treating the surfaces with dopamine before applying the coating [87]. This nanocomposite showed reduced bacterial adhesion and enhanced corrosion resistance against *Escherichia coli* and *Staphylococcus aureus*. Yoon and colleagues [116] conducted experiments with superhydrophobic and superhydrophilic layers on steel, using carbon nanotubes–polytetrafluoroethylene (CNT–PTFE) and TiO_2_, respectively. They tested these surfaces with *Escherichia coli* suspensions under varying flow rates. The surface morphologies of these nanocomposites were analyzed using field emission scanning electron microscopy (FESEM) and atomic force microscopy (AFM), contributing valuable insights into their structural and functional properties.

### 2.3. Anti-Biofouling Polymer Coatings on the Steel

Antifouling surfaces are designed not to deactivate bacteria but to hinder or substantially diminish the adherence of bacteria, thus impeding biofilm development. This approach effectively curtails the growth of bacterial colonies on surfaces that encounter food, thereby reducing the likelihood of contamination and cross-contamination. The primary goal of anti-biofouling or anti-adhesive surfaces is to deter the initial adherence of microorganisms, thereby obstructing the formation of stable biofilms through various surface modification techniques.

Two principal methodologies for creating anti-biofouling (anti-adhesive) surfaces include the development of superhydrophobic surfaces on steel and the application of repulsion-based antifouling coatings. Superhydrophobic approaches leverage microscale and nanoscale surface roughness or porosity, coupled with coatings that are low in surface energy, to create air pockets (depicted as white) that physically block bacterial contact and adherence. Conversely, repulsion-based methods utilize dense layers of flexible polymer chains to repel bacteria, thus preserving their viability.

Creating a durable superhydrophobic surface on steel remains a challenge, largely due to weak adhesion between the coating and the substrate and the coating’s instability under typical steel operating conditions [117]. Several techniques and strategies have been employed to induce superhydrophobicity on steel, predominantly through surface treatment with low-surface-energy materials such as long perfluorinated chain silane and various texturing additives like CuS [118], Boehmite alumina [119], nickel [117], and others. In the fabrication of superhydrophobic polymer coatings on steel, one approach involves depositing materials like electroplated nickel and electrodeposited polymer composite materials that include nickel and carbon nanotubes. The development of multilayer hydrophobic polymer coatings such as polytetrafluoroethylene (PTFE), electroless nickel-PTFE (EN-PTFE) [120], and polymeric nano-composite coatings is also a noteworthy method [116,121,122,123].

Electrodeposition of conductive polymers presents a technique for the precise manipulation of surface topography. In [124], superhydrophobic surfaces were engineered via the electrodeposition of hydrophobic polymers (PEDOT-F4 or PEDOT-H8) onto steel, enabling control over surface texture. The findings indicated that achieving anti-bio-adhesive and anti-biofilm characteristics hinges on managing surface topographical elements, combined with ensuring minimal water adherence (Cassie–Baxter state) and limiting crevice formation at the bacterial cell scale (nano-scale structures).

A separate investigation [116] developed superhydrophobic layers on steel by annealing the metal with carbon nanotubes-polytetrafluoroethylene (CNT–PTFE) and titanium dioxide (TiO_2_). The CNT–PTFE superhydrophobic coating exhibited minimal bacterial adherence, attributed to its lotus-like effect.

Ni-P-PTFE coatings are recognized for their exceptional non-stick and low-friction properties. Extensive research [125,126] has explored the use of these antifouling coatings in reducing fouling during food processing, however, concerns have been raised by the FDA regarding potential food contamination due to coating instability on food equipment surfaces. Chemically, antifouling coatings such as Ni-P-PTFE, which combines polytetrafluoroethylene (PTFE) with nickel and phosphorus, are applied to equipment surfaces. It was demonstrated in [125] that Ni-P-PTFE coatings possess lower wear resistance and adhesion to steel and significantly reduce *Escherichia coli* adherence by about 90%. Identifying a coating material that bonds effectively with steel and minimizes fouling is crucial. For example, Ni–PTFE modified steel surfaces in dairy processing have been observed to lessen milk and bacterial fouling by over 96% [127,128]. Despite their effectiveness, these PTFE anti-adhesive coatings can be costly, require complex application methods, and potentially release harmful substances like perfluorooctanoic acid. A novel approach using graded electroless Ni–P–PTFE coating has been shown to decrease bacterial attachment by 82–97% [129]. A novel approach utilizes nickel-graphene oxide (Ni-GO) and nickel-reduced graphene oxide (Ni-rGO) composites for their dual antibacterial and anti-corrosive capabilities, specifically targeting *Staphylococcus aureus*. This method leverages magnetic field-assisted scanning jet electrodeposition to apply the active compounds onto manganese steel, crafting surfaces that are not only resistant to corrosion but also possess potent antibacterial effectiveness [130].

Surfaces engineered for repulsion-based antifouling capitalize on the natural tendency to repel bacterial cells away from their substrates. This characteristic impedes the adhesion and subsequent proliferation of bacteria. Echoing the characteristics of the previously mentioned superhydrophobic surfaces, these repulsion-centric surfaces also inhibit bacterial adhesion but lack bactericidal properties. However, unlike their superhydrophobic counterparts, these surfaces are characteristically hydrophilic. Techniques in anti-biofouling via repulsion involve reducing surface energy, employing antifouling coatings like self-assembled monolayers with hydrophilic polymer brushes such as poly(ethylene glycol) (PEG) [131], attaching dopamine with PEG ends, employing cold plasma or silane coupling agents for PEG grafting, and anchoring lysozymes or PEG to poly(ethylene imine) coated substrates. Although PEG stands as the most prevalent substance in these applications, alternatives such as polyacrylates, polyamides, and polysaccharides have also proven to be efficacious. In the dairy industry, PEG has been used to develop brush polymer coatings on equipment to combat both bacterial and protein fouling. Research by Zouaghi et al. [132] has successfully utilized steel substrates to create surfaces with impressive antifouling properties, particularly demonstrated in pilot-scale milk pasteurization processes. Another study [133] highlighted the success of repulsion-focused antifouling through the application of radio frequency (RF) plasma polymerization (PlzP) employing hydrophilic monomers such as Polyethylene glycol and Polyhydroxyethylmethacrylate on stainless steel (SS 316) surfaces. This treatment notably decreased *Enterobacter sakazakii* attachment by 99.74% compared to unmodified surfaces. Poly(ethylene oxide) (PEO), synonymous with poly(ethylene glycol), is known for its exceptional protein fouling resistance. Directly applying PEO to steel in dairy processing is challenging, thus necessitating PEO-incorporated coatings. Studies have indicated that silicone coatings modified with PEO, created through bulk modification using PEO-silane amphiphiles with short siloxane linkages, are remarkably resistant to both plasma proteins and bacterial adhesion.

In work by Terada et al. [134] a polyethylene sheet was modified with glycidyl methacrylate (GMA) and subsequently converted to a negatively charged surface using sodium sulfite, significantly reducing *Escherichia coli* adhesion and altering biofilm structures. In the study [135], steel was treated with a 1% Nafion coating, a sulfonated tetrafluoroethylene-based fluoropolymer-copolymer known for its ionic, thermal stability, and biocompatibility, and showed significant reductions in *Escherichia coli* adhesion. The effectiveness of these coatings, attributed to electrostatic repulsion, is due to the substantial number of sulfonate groups in Nafion polymers. In another study [136], SS-azide surfaces underwent treatment with both antifouling and antibacterial polymer brushes: alkyne-functionalized poly(N-hydroxyethylacrylamide) (alkynyl-PHEAA) for preventing fouling, and alkyne-functionalized poly(2-(methacryloyloxy)ethyl trimethylammonium chloride) (alkynyl-PMETA) for combating bacteria. The effectiveness of these polymer-functionalized surfaces was demonstrated through reduced adsorption of bovine serum albumin and lower bacterial fouling by Gram-negative *Escherichia coli* and Gram-positive *Staphylococcus epidermidis*. A summary of the approaches to imparting antibacterial properties to steel surfaces is given in Table 3.

## 3. Conclusions

The quest to thwart the growth of bacterial biofilms on the metal surface in the food industry is a significant endeavor in scientific research. Antibacterial polymer coatings, through mechanisms such as controlled release, direct contact killing, and imparting antibiofouling characteristics to steel surfaces, achieve a broad and potent suppression of foodborne pathogens. The application of antimicrobial polymer coatings can solve a number of important problems with improving the safety of technology to work with food, and can also improve the corrosion and decorative properties of metal.

In this review, various critical aspects related to bactericidal polymer coatings for steel surfaces, highlighting their mechanisms, applications, and limitations in combatting microbial fouling were observed. The discussion includes:-An overview of how bactericidal polymer coatings are classified according to their interaction mechanisms with microbes. This classification helps in understanding the diverse strategies employed to prevent microbial colonization.-In-depth analysis of different strategies to achieve antibacterial properties: released- and contact-based antimicrobial coatings and antibiofouling strategies for steel coatings.-The impact of antimicrobial additives on the mechanical and corrosion resistance properties of coatings is significant. This factor necessitates careful consideration in selecting the appropriate bactericidal strategy for real-world applications.

## Figures and Tables

**Figure 1 polymers-16-00809-f001:**
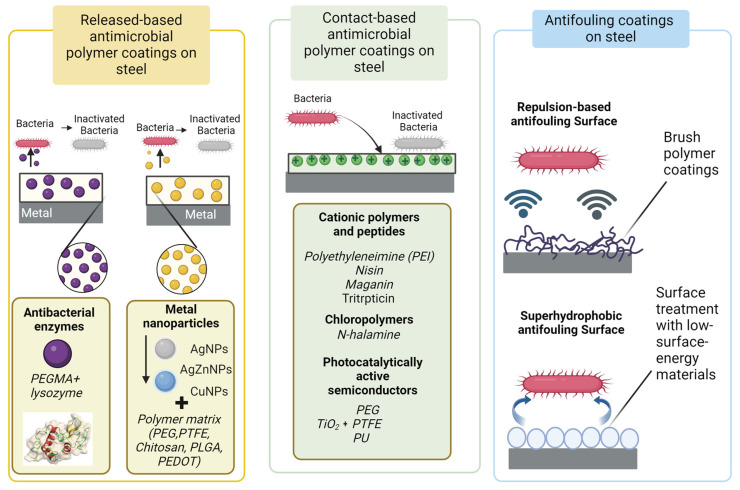
Antimicrobial polymer coating strategies on steel surface.

**Table 2 polymers-16-00809-t002:** Released-based antimicrobial polymer coatings with antimicrobial peptides on steel.

Substrate	Peptide	Polymer Coupling Agent	Antibacterial Activity Species Tested	Reference
Steel	Magainin 1 Nisin	Chitosan polymer coating	*Listeria ivanovii*	[79]
Steel	Nisin Tritrpticin (Trp11) Palmitoyl-4K (4K-C16)	Epoxy polymer coating	*Bacillus subtilis* *Escherichia coli*	[97]
Steel	Magainin II	Dopamine	*Staphylococcus aureus* *Escherichia coli*	[98]
Steel	Nisin	Methacrylamide bearing (oxidized) 3,4-dihydroxyphenylalanine (mDOPA	*Bacillus subtilis*	[80]

**Table 3 polymers-16-00809-t003:** Comparative analysis of approaches for imparting antibacterial properties to steel surfaces.

Antibacterial Coating Strategy	Working Agents	Mechanism of Action and Advantages	Limitations
Released-based antimicrobial coating	Antimicrobial-loaded (metals NPs, antibiotics enzymes)	Simple release. Broad-ranging effectiveness, with the capability to deliver a substantial quantity of antimicrobial substance.	Impact of bacterial suppression is momentarily constrained by the available stock of antimicrobial agents. Potential toxicity from the biocidal substance. Risk of prompting bacterial immunity. Diminution of the antimicrobial compound. Unselective diffusion of antimicrobial elements.
Contact-based antimicrobial coating	Cationic polymers and peptides, N-halamine, photocatalytic sensitive compounds (TiO_2_)	Cellular interference upon interaction with an active compound Possibility for sustained operational effectiveness	Activity confined to the vicinity of the altered surface. Diminished effectiveness upon contact with the body. Reduced efficacy of the photocatalytic contact-killing layer under ambient lighting conditions.
Anti-biofouling coating	EN-PTFE, PEDOT, CNT-PTFE, Ni-P-PTFE, PEG, PEO, alkynyl-PMETA, alkynyl-PHEAA	Deterring bacteria through alterations in surface energy Mechanisms that are non-toxic to cells Initial prevention of bacterial colonization at the onset of contamination	Activity limited to the treated surface. Absence of bactericidal effect Low stability of the surface properties

## Data Availability

Not applicable.

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
