# Peer review of "A Review of Antimicrobial Polymer Coatings on Steel for the Food Processing Industry"

_polymers, 2024, doi:10.3390/polym16060809_

Round 1

Reviewer 1 Report

Comments and Suggestions for Authors

The authors reviewed the antimicrobial polymer coatings for steel surfaces. The review is well-organized and provides a good summary for readers. Here are some suggestions:

1.     The title should be revised. The word ‘laminating’ was only used in your title. Please make the title more related to your content and easier to be understand.

2.     Check the format for the latin name of the microorganisms. It should be italic, such as line 40-41 and line 74.

3.     Check the format of TiO2 in section 2.2.3.

4.     The authors mentioned that food contact materials play a crucial role in food safety. However, they should discuss it more and refer to some previous studies to point out the food safety of specific antimicrobials.

5.     The iThenticate report showed a very high replicate percentage of your review, you should carefully revise it. 

Author Response

Thank you very much for taking the time to review this manuscript. Please find the detailed responses below and the corresponding corrections and highlighted changes in the re-submitted files.

Comments 1: The title should be revised. The word ‘laminating’ was only used in your title. Please make the title more related to your content and easier to be understand.

Response 1: In accordance with your comments, the title of the article has been corrected to: Antimicrobial polymer coatings on steel for food field.

Comments 2: Check the format for the latin name of the microorganisms. It should be italic, such as line 40-41 and line 74?

Response 2: All names of the microorganisms mentioned in the article have been standardized in their formatting.

Comments 3: Check the format of TiO2 in section 2.2.3.

Response 3: This change was made to the text (section 2.2.3)

Comments 4: The authors mentioned that food contact materials play a crucial role in food safety. However, they should discuss it more and refer to some previous studies to point out the food safety of specific antimicrobials.

Response 4: The topics of food safety and food contamination are separate, very extensive directions in science, and there are review articles from recent years dedicated specifically to this topic, to which references have been added in the text of this review in accordance with your recommendation.

Comments 5: The iThenticate report showed a very high replicate percentage of your review, you should carefully revise it

Response 5:  The text of the article has been significantly revised in accordance with your comments. Changes are highlighted in yellow in the text. However, given that the topic is quite narrow and there are not many works dedicated specifically to polymer antibacterial coatings on steel for food application, there may be overlaps with other review works in terms of the literary sources used.

Reviewer 2 Report

Comments and Suggestions for Authors

Dear Authors,

Your review article is interesting and well organized with suitable sections/subheadings.

However, the title could be composed in a better way. A suggestion is given in the attached copy of your article. Abstract needs a better editing. Please check it again.

Also, in the title as well as in the abstract you are mentioning that these coatings are helpful for steel surfaces in the healthcare sector as well, but there is no cases/examples for this sector, only for food sector are provided examples/description. Please modify these sections.

In the Introduction you need to make a better review of the literature with newer references beside those that are already used. Steel material is described in a certain content, but there is not given what are the most used types of stainless steel in the food sector, for example. Microbial organisms should be properly written e.g. with Italic letters, Gram-positive or so is written with Capital or small letters (there are different styles used in this paper).

Other sections are suitable proposed, but suggestions for improvement are given in the attached copy.

Conclusion part could be modified in terms of more concise and clear descriptive way.

Thank you.

Comments on the Quality of English Language

English could be improved.

Author Response

Thank you very much for taking the time to review this manuscript. Please find the detailed responses below and the corresponding corrections and highlighted changes in the resubmitted files.

Comments 1:  However, the title could be composed in a better way. A suggestion is given in the attached copy of your article. Abstract needs a better editing. Please check it again.

Response 1: In accordance with your comments, the title of the article has been corrected to: Antimicrobial polymer coatings on steel for food field. The abstract has been formatted and improved.

Comments 2:  Also, in the title as well as in the abstract you are mentioning that these coatings are helpful for steel surfaces in the healthcare sector as well, but there is no cases/examples for this sector, only for food sector are provided examples/description. Please modify these sections

Response 2: The word "medical" was introduced into the title because certain types of coatings may be applied or are already being used in this field. However, as you correctly noted that the main focus of the article is specifically on food contact coatings, we have edited the title of the article in accordance with your recommendations.

Comments 3: In the Introduction you need to make a better review of the literature with newer references beside those that are already used. Steel material is described in a certain content, but there is not given what are the most used types of stainless steel in the food sector, for example. Microbial organisms should be properly written e.g. with Italic letters, Gram-positive or so is written with Capital or small letters (there are different styles used in this paper).

Response 3: In the introduction part, references to new articles have been added, as well as a discussion on the types of steel that are most commonly used in the food sector. All names of the microorganisms mentioned in the article have been standardized in their formatting.

Comments 4: Other sections are suitable proposed, but suggestions for improvement are given in the attached copy.

Response 4: Corrections have been made to the text in accordance with the suggestions for improvement attached to the copy. Changes are highlighted in yellow in the text.

Comments 5: Conclusion part could be modified in terms of more concise and clear descriptive way.

Response 5: Conclusion part has been rewritten and made more concise. 

Round 2

Reviewer 2 Report

Comments and Suggestions for Authors

Dear Authors

Your second version of the manuscript is improved compared to the previous one.

The title needs a change in order to be more attractive and have a better ending. Some suggestions are included in the attached version of your manuscript.

Abstract is improved and more focussed now, still some clarifications are needed.

The Introduction part is very good/improved in general as well as theoretical review of variety of coatings that exist nowadays.

good work.

The Conclusion part is shorten and more focussed now. 

Thank you.

Comments on the Quality of English Language

it could be improved.

Author Response

Dear reviewer, thank you for your comments. We have tried to take them all into account and have made the corresponding changes to the text of the article. The title of the article has been changed in accordance with your suggestion. Changes have been made to the abstract section (highlighted in green in the text). Changes have also been made to the main body (additional references added) in accordance with your comments.
